Trials in developing a nanoscale material for extravascular contrast-enhanced ultrasound targeting hepatocellular carcinoma

Wu Size wsz074@aliyun.com 1
Lin Xiyuan 2
1 Department of Ultrasound, The First Affiliated Hospital of Hainan Medical University , Haikou , Hainan Province , China
2 Department of Emergency Medicine, The First Affiliated Hospital of Hainan Medical University , Haikou , Hainan Province , China
Marunaka Yoshinori
Electronic publication date: 2020 Dec 7
Publication date: 2020
Volume: 8
Electronic Location ID: e10403
Received 2020 Aug 12; Accepted 2020 Oct 31
Copyright: ©2020 Wu and Lin
Copyright year: 2020
Copyright holder: Wu and Lin
License: This is an open access article distributed under the terms of the Creative Commons Attribution License, which permits unrestricted use, distribution, reproduction and adaptation in any medium and for any purpose provided that it is properly attributed. For attribution, the original author(s), title, publication source (PeerJ) and either DOI or URL of the article must be cited.
License URL: https://creativecommons.org/licenses/by/4.0/

Keywords: Contrast agent, Ultrasound, Extravascular imaging, Target, Hepatocellular carcinoma, Negative outcome

Funding: National Natural Science Foundation of China 81560290 This project was supported by the National Natural Science Foundation of China (Grant No. 81560290). The funders had no role in study design, data collection and analysis, decision to publish, or preparation of the manuscript.

==============================
Background

Medical imaging is an important approach for the diagnosis of hepatocellular carcinoma (HCC), a common life threaten disease, however, the diagnostic efficiency is still not optimal. Developing a novel method to improve diagnosis is necessary. The aim of this project was to formulate a material that can combine with GPC3 of HCC for targeted enhanced ultrasound.

Methods

A material of sulfur hexafluoride (SF6) filled liposome microbubbles and conjugated with synthesized peptide (LSPMbs) was prepared and assessed in vitro and vivo. Liposome microbubbles were made of DPPC, DPPG, DSPE-PEG2000,and SF6, using thin film method to form shell, followed filling SF6, and conjugating peptide. A carbodiimide method was used for covalent conjugation of peptide to LSMbs.

Results

The prepared LSPMbs appeared round shaped, with size of 380.9 ±  176.5 nm, and Zeta potential of −51.4 ±  10.4mV. LSPMbs showed high affinity to Huh-7 cells in vitro, presented good enhanced ultrasound effects, did not show cytotoxicity, and did not exhibit targeted fluorescence and enhanced ultrasound in animal xenograft tumors.

Conclusion

Extravascular contrast-enhanced ultrasound targeted GPC3 on HCC may not be realized, and the reason may be that targeted contrast agents of microbubbles are hard to access and accumulate in the tumor stroma and matrix.

Introduction

Hepatocellular carcinoma (HCC) is a common primary malignant neoplasm derived from hepatocytes, especially in some Asia-Pacific regions, where the underlying diseases of hepatitis B virus infection and relevant diseases are in high prevalence (Zhu et al., 2016b). At present, the diagnosis of small and atypical HCC is still challenging (Cassinotto, Aubé & Dohan, 2017). Healthcare professionals and scientists have been searching new methods to improve diagnosis efficiency. Targeted imaging has been a heat topic and interest of researchers in recent decades, and is expected to be an ideal non-invasive imaging method (Ofuji et al., 2014; Zhu et al., 2016a; Wu, 2017). Although the lack of a basement membrane and smooth muscle and the expansion of the intercellular space in cancer vasculature result in a maximum pore size of approximately 380-780 nm, which exhibits leaky and/or defective blood vessels, microbubbles (Mbs) with diameter more than 1,000 nm cannot migrate from the tumor vasculature to the cellular target site to exert the desired diagnostic effect (Ofuji et al., 2014). Therefore, the development of nanoscale targeted ultrasound contrast agents (UCAs), which may permeate through the tumor vasculature gap and bind to tumor cells, with extravascular imaging function, is required. On the HCC cellular membrane, there is a high expression of Glypican-3 (GPC3) protein, which can be used for a target for molecular imaging (Zhu et al., 2016a; Wu, 2017). However, because of the antigenicity and larger size of GPC3, if it is used for the ligand of a targeted material, the material may not produce desired effect in the vivo. On this condition, if a small size peptide without antigenicity but possessing similar targeting ability, it may be used for the fabrication of a targeted contrast material. We hypothesized that a new material may be fabricated, with function of targeted contrast-enhanced ultrasound (CEU) imaging. Based on previous studies, we established a sort of liposome microbubbles and conjugated with a synthesized peptide targeting GPC3 of the HCC, with liposome as shell and sulfur hexafluoride gas (SF6) as the core (Zhu et al., 2016a; Wu, 2017).

Materials & Methods

Ethics statement

All the experimental procedures in this study were in compliance with the National Institutes of Health guidelines and were approved by the Institutional Animal Care and Use Committee of Hainan Medical University (2018-02-27).

Preparation of liposomes and liposome microbubbles

Sulfur hexafluoride gas (SF6) filled liposome microbubbles and conjugated with synthesized peptide (LSPMbs), SF6 filled liposome microbubbles and conjugated with GPC3 antibody (LSGMbs), SF6 filled liposome microbubbles and conjugated with synthesized peptide, and liposomes (LS) were prepared and assessed in the public scientific research center of Hainan medical university. To develop a GPC3 targeted liposome microbubbles, materials and formula were chosen and optimized, and the optimum formula was selected through orthogonal design test according to the enhanced ultrasound effect. 1,2-dipalmitoyl-sn-glycero-3-phosphocholine (DPPC), 1,2-Dipalmitoyl-sn-glycero-3-phospho-sn-1-glycerol (DPPG), 1,2-distearoyl-sn-glycero-3-phosphoethanolamine-N-[carboxy(polyethylene glycol)-2000] (DSPE-PEG2000) , GPC3, artificial synthesized peptide of DHLASLWWGTEL and SF6 were adopted and fabricated using following protocols. 12  mg DPPC was dissolved in 3.0  mL chloroform, followed adding 1.5  mL methanol and 0.5 mL deionized water to form a mixture, next dissolved 23 mg DPPG and 11 mg DSPE-PEG2000 in the mixture. The solvent was removed by rotary evaporation and vacuum using a Vacuum Rotary Evaporator (Xian Depai17 Biotechnology Co., Ltd. Xian, China), and the mixture formed a thin film on the wall of the container. 20 mL deionized water was added to the container to harvest and suspend the lipid mixture, next it was transferred to a tube to undergo magnetic stirring at 65 °C for 20 min, followed by probe-sonication (Vibracell™, VCX130PB/VCX130 Sonics, Sonics and Materials, Inc., Newtown, USA) with a frequency of 40 kHz under 35 °C f or 5 min. A mechanical blender (Ultra-turrax T25, Janke & Kunkel, IKA-Labortechnik, Staufen, Germany) was used for liposome microbubbles preparation. The above processed lipid dispersion was put into a 50  mL Falcon tube, the air above the aqueous dispersion in the tube was replaced with SF6 gas, and the tube was sealed with parafilm. The temperature of lipid dispersion was heated to 30 °C , the homogenizer was operated to create high shear mixing (15,000 rpm, 5 min) to form microbubbles. The mixture was centrifuged at 12,000 rpm at 4 °C for 5 min, washed with deionized water, three times. 15% (w/v) sucrose solution was added to the mixture in a 5:1 volume ratio (mixture: sucrose), small glass vial of four  mL volume was used for the loading, each with two  mL mixture. The air in the vials was replaced with SF6 gas before lyophilized in a −85 °C lyophilizer (SP Scientific, VirTis, USA). After 24 h completely freeze-drying, vials were refilled with SF6 gas and sealed, stored at 4 °C . Pure liposomes (LS) were prepared using the above protocols without filling SF6 gas.

Preparation of GPC3 targeted liposome microbubbles

Firstly, after liposome microbubbles (LSMbs) were prepared using the protocols above, but did not add sucrose solution and lyophilize. Next, a carbodiimide method was used for covalent conjugation of the synthesized peptide to the free carboxyl groups on the surface of LSMbs. The prepared LSMbs were resuspended in MES buffer (0.1 mol/L, pH 6.0), and an adequate amount of EDC/NHS [1-ethyl-3(3-dimethylaminopropyl) carbodiimide (EDC): N-hydroxy succinimide (NHS) = 1:4, Sigma-Aldrich Chemical Co., Inc, USA] were added into the suspension. The mixture suspension was oscillated and incubated for 2 h at room temperature (25 °C). The remaining EDC/NHS was removed by three-time centrifugation at 12,000 rpm using MES (pH 6.0), 5 min each time. The precipitate was dispersed into MES buffer (0.1 mol/L, pH 8.0), and an adequate amount of synthesized peptide was added and incubated with stirring for 2 h at room temperature. The peptide was compounded by GL Biochem (Shanghai) Ltd. (Shanghai, China) in accordance to a 12-mer peptide with the sequence of DHLASLWWGTEL reported from previous study that it can target GPC3 of HepG2 (Zhu et al., 2016a) (DOI: 10.1021/acs.bioconjchem.6b00030). The mixture was centrifuged at 12,000 rpm at 4 °C for 5 min, washed by deionized water, three times. Next, 15% (w/v) sucrose solution was added to the mixture in a 5:1 volume ratio (mixture: sucrose), small glass vial of four  mL volume was used for the loading, each with two  mL mixture. The air in the vials was replaced with SF6 gas before lyophilized in a −85 °C lyophilizer (SP Scientific, VirTis, USA). After 24 h completely freeze-drying, vials were refilled with SF6 gas and sealed, stored at 4 °C. Liposome microbubbles (LSMbs), and GPC3 antibody (BM1846; Wuhan Boster Biological Technology, Ltd., Wuhan, China) conjugated liposome microbubbles (LSGMbs) were prepared in the similar protocol above.

Characterization of LSPMbs

Transmission electron microscopy evaluation

The LSPMbs were observed using transmission electron microscope (TEM) for particle size and shape assessment. LSPMbs were suspended with deionized water (1:50), and one drop of the suspension was dropped onto a carbon-coated copper grid. After drying and adhesion in 25 °C , samples were negatively stained by sodium phosphotungstate solution (2%, w/w) and analyzed with a 120-kV TEM (TEM; JEM 2100, JEOL, Tokyo, Japan). Suspensions of samples with different concentration were dropped on coverslips and observed under light microscopy.

LSPMbs Size and zeta potential measurements

LSPMbs size and Zeta potential of each sample was measured using a Zetasizer Nano S90 (Malvern Instruments Ltd., Malvern, UK) by Laser Doppler Anemometry (LDA) using electropheoretic light scattering at 25 °C . An adequate amount of LSPMbs was suspended and diluted to enable the microbubbles concentration was maintained to ensure that multiple scattering and microbubble–microbubble interactions were negligible. The microbubble size and zeta potential of each sample were measured three times, and the mean value was taken as the final microbubble size and zeta potential. LSGMbs were assessed in the same methods.

Assessment of biocompatibility of LSPMbs

MTT assay for cytotoxicity

RAW 264.7 cells (Cell bank of the Chinese academy of sciences (Shanghai, China)) in exponential phase of growth were taken, transferred 200 µL to each well of a 96-well plate, and adjust the cell density to 5000/well. The cells were cultured with DMEM (Wuhan Boster Biological Technology, Ltd., Wuhan, China), 5% fetal bovine serum (FBS) (Gibco, Australia), and 1% penicillin-streptomycin at 37 °C in 5% CO2 atmosphere for 24 h, when the well were fully covered, added different concentration gradients LSPMbs (2 mg/mL, 5 mg/mL, 10 mg/mL, 15 mg/mL, and 20 mg/mL) prepared using with DMEM and 5% FBS to different wells, 200 µL per well, then continuing incubated for 24 h. LSPMbs were burst using ultrasound at the experiment. Next, 10 µL 0.5% 3-(4,5-dimethylthiazol-2-yl)-2,5-diphenyl-2H-tetrazolium bromide (MTT) solution was added to each well, and continuing incubated for 24 h, 48 h and 72 h in the dark place, respectively. The incubation was ended at 24 h, 48 h and 72 h, respectively, and the solution in the wells were carefully absorbed out and discarded. Next, 200 µL dimethyl sulfoxide was added to each well, and oscillated at a frequency of 20 times per minute for 10min to fully dissolve the crystal. Optical density (OD) is measured at the 570 nm wavelength by using a spectrophotometric microplate reader (Bio-Tek ELX-800, Winooski, VT, USA). Controls were established and processed using the similar protocols.

Evaluation of enhanced ultrasound imaging of LSPMbs in vitro

To assess the contrast-enhanced effect of LSPMbs, LSPMbs were suspended with deionized water and diluted into various concentrations (1.6, 0.8, and 0.4  mg/mL) and placed in different plastic tubes for ultrasound evaluation. LSPMbs suspension filled tubes were fixed in a water container, and their CEU effect was assessed using GE Logiq E9 ultrasound system (GE Healthcare, Milwaukee, WI, USA), using a  mL6-15-D linear transducer with a frequency of 4–15 MHz. During the ultrasound performance, the frequency of the transducer was set to 12 MHz, the depth and focus were adjusted to optimize imaging, and the model was shifted to contrast imaging, using the default parameter (MI 0.1). Before the ultrasound scanning, the tubes were agitated slightly. Controls of commercial ultrasound contrast agent SonoVue (Shanghai Bracco Sine Pharmaceutical Corp. Ltd., Shanghai, China), degas deionized water, and air were established, and these were assessed using the same protocols above.

Enhancement ultrasound imaging effects of LSPMbs, LSGMbs, SonoVue, LS, degas deionized water, and air were determined using Photoshop software (Adobe Photoshop CS3, Adobe Systems Inc, CA, USA). To analyze ultrasound images, observers opened the image in Photoshop, activated menus of “Window, Information, and Histogram” consecutively, selected “rectangle, and statistics display ” tools, set the same region of interest to the tube, parameters yielded automatically, measured three times in each image, and adopted the mean value of scales (arbitrary units) as the result of a single image.

Assessment of affinity of LSPMbs to the liver cancer cells

Fluorescence experiment was used for the assessment of affinity of LSPMbs to the liver cancer cells. A tiny amount of 1,1′-dioctadecyl-3,3,3′,3′-tetramethylindocarbocyanine perchlorate (DiI) (Yeasen Biotech Co., Ltd., Shanghai, China) was added into five  mL of 16  mg LSPMbs suspension to form DiI labelled LSPMbs suspension. Huh-7 cells (Cell bank of the Chinese academy of sciences (Shanghai, China)) were seeded in a six-well plate with one glass slip placed in each well at a concentration of 3 ×104 cells/well. Next day, the Huh-7 cells were fixed by 90% cold ethanol for 20 min and blocked by 10% bovine serum albumin (BSA) at 37 °C for 1 h and subsequently incubated with 4 drops of DiI labelled LSPMbs suspension for 3 h in dark place. Next, the wells and slips were washed with PBS three times, and the cells on the slips were mounted with 4′, 6-diamidino-2-phenylindole (DAPI, Wuhan Boster Biological Technology, Ltd., China) for nuclei visualization and detected using a laser confocal microscope (Fluoview FV 10001000, Olympus, Japan). Images of bright, DAPI staining, DiI staining, and merged were obtained. LSGMbs were assessed in the same protocols for control. To confirm the specificity of binding of DiI labelled LSPMbs to GPC3 in the Huh-7 cells, 0.1 mL synthesized peptide (10 µg/mL) was applied before adding DiI labelled LSPMbs as a blocking control, and the other protocols were the same as the above.

Assessment of LSPMbs in vivo

All animal experimental procedures were approved by the Hainan Medical University Association for Accreditation of Laboratory Animal Care. Animals of female BALB/C mice were used for the evaluation of targeting ability and contrast-enhanced effect. Twenty Huh-7 cell xenograft tumor models of female BALB/C mice were established, and the Huh-7 cell line was acquired from the cell bank of the Chinese academy of sciences (Shanghai, China). All animals expect four mice were sacrificed by euthanasia using isoflurane after fluorescent imaging were sacrificed using carbon dioxide in a closed box at the end of the animal experiments. The criteria of animal death are that the mice were in collapse state, losing muscular tension, no breath, no heart beating, and the skin color became gray.

Enhanced ultrasound imaging assessment of LSPMbs

Fifteen of the mice were used for the targeted CEU experiments in five groups, with each group of three mice. The CEU effect was assessed using GE Logiq E9 ultrasound system (as has addressed in the previous section). During the ultrasound scanning, the frequency of the transducer was set to 12 MHz, the depth and focus were adjusted to optimize imaging, and the model was shifted to contrast imaging, using the default parameter (MI 0.1). The shape of the xenograft tumors was ovoid, and the longitudinal diameter of the tumors of 24 mice was 10.4 ± 0.53 mm in the fifth week after cells seeded. Control experiments were conducted in four groups using LSGMbs, LSPMbs (GPC3 antibody or synthesized peptide blocked previously), SonoVue (a commercial ultrasound contrast-enhanced agent), and LS, respectively. LSPMbs suspension was prepared using 14 mg LSPMbs and four mL 0.9% sodium chloride solution, agitated slightly before injection. The animals with Huh-7 xenograft tumor in five groups were intravenously injected suspension of LSPMbs, LSGMbs, LSPMbs (injected 0.1 mL synthesized peptide or 0.1 mL dilated GPC3 antibody for blocking in 3 min ahead), SonoVue (15  mg SonoVue in four  mL 0.9% sodium chloride solution), and LS (14  mg LS and four  mL 0.9% sodium chloride solution, agitated slightly before injection), respectively, each with the volume of 0.2  mL. The contrast imaging time was counted since the bolus intravenous injection of 0.2 mL LSPMbs suspension via the mouse tail vein. The images were saved in the ultrasound system and exported for later study.

Enhancement ultrasound imaging effects of LSPMbs, LSGMbs after synthesized peptide or GPC3 antibody blocking, SonoVue, LS were determined using Photoshop software, and the methods had been addressed in the previous section.

Fluorescent imaging assessment of LSPMbs

Ten mice with xenograft tumors of Huh-7 cells allotted to two groups of five each were used to conduct Cy 5.5 fluorescence experiment to test the specificity and affinity of the peptide in LSPMbs to GPC3 of the liver cancer. An IVIS Lumina image system (Xenogen) (IVIS® Lumina XR) (Caliper life sciences) was used for the evaluation. During the fluorescence imaging, mice were under gas anesthesia with oxygen and isoflurane (Jinan Shengqi pharmaceutical Co, Ltd., Jinan, China). 0.2 mL Cy5.5 labelled LSPMbs suspension was intravenously injected into six mice with Huh-7 xenograft tumor via the tail vein, images were acquired at one minute, six hours, and 24 h. Cy5.5 labelled LSPMbs suspension was prepared using 14 mg Cy5.5 labelled LSPMbs and four  mL 0.9% sodium chloride solution, agitated slightly before injection. The control experiments were conducted in five mice with Huh-7 xenograft tumors, with the same methods after injection of 0.1 mL synthesized peptide (10 µg/mL) for blocking in three minutes. Four mice were sacrificed by euthanasia using 3 mL isoflurane in a closed small box. The tumor, liver, heart, lung, kidney, and spleen of the mice were isolated for fluorescent imaging assessment at 24 h.

Statistical analysis

Quantitative data are presented as mean ± SD (standard deviation) , and qualitative data are presented as percentile. Statistical significance of differences between groups of quantitative variables were analyzed using paired-sample t tests or univariate analysis of variance, and qualitative variables were analyzed using Chi-square test. All statistical analyses were performed using SPSS software (Version 20; IBM, Armonk, NY, USA). P < 0.05 was considered significant.

Results

Characterization of LSPMbs

The LSPMbs appeared round shaped on a sectional view, with different size, without aggregation, identified by transmission electron microscopy (Fig. 1).

Figure 1 Image of the LSPMbs obtained by transmission electron microscope.

Image of the LSPMbs obtained by transmission electron microscope. The LSPMbs present round shaped on a sectional view, with different size, without aggregation.

Size and Zeta potential of LSPMbs were 380.9 ± 176.5 nm and −51.4 ± 10.4 mV, respectively. The determination results showed “Good” (Figs. 2 and 3).

Figure 2 Size and distribution of LSPMbs.

Size of LSPMbs are 380.9 ± 176.5 nm, and the determination results show “Good” quality.

Figure 3 Zea potential and distribution of LSPMbs.

Zea potential of LSPMbs was −51.4 ± 10.4 mV, and the determination results show “Good” quality.

Assessment of biocompatibility of LSPMbs

MTT assay for cytotoxicity

The cell index had no significant differences among different LSPMbs concentrations at different time (all P >  0. 05), indicating that LSPMbs did not cause significant toxic effect on RAW 264.7 cells. As shown on Fig. 4.

Figure 4 MTT assay for cytotoxicity showed the cell index had no significant differences among different LSPMbs concentrations at different time (all P> 0. 05).

Evaluation of enhanced ultrasound imaging of LSPMbs in vitro

LSPMbs and SonoVue suspension with different concentrations presented different enhancement effects, they presented almost the same enhancement effect at the same concentration (Figs. 5A and 5D were obtained from 1.6 mg/mL, Figs. 5B and 5E was obtained from 0.8 mg/mL, and Figs. 5C and 5F was obtained from 0.4 mg/mL). At higher concentration (1.6 mg/mL), they all presented homogeneous hyperechogenicity with marked attenuation (Figs. 5A and 5D), and the echogenicities became weaker when the concentrations decreased (Figs. 5B and 5E, and Figs. 5C and 5F), and they all much stronger than control of degas deionized water, which presented homogeneous anechogenicity (Fig. 5G). Control of air presented strong echogenicity at the interface of the tube, and the distal field presented marked attenuation (Fig. 5H), which was substantial different from those obtained from LSPMbs and SonoVue suspension.

Figure 5 Ultrasound images of the LSPMbs, SonoVue, degas deionized water, and air in the tubes.

(A & D) were obtained from LSPMbs and SonoVue of 1.6 mg/mL, B and E were obtained from LSPMbs and SonoVue of 0.8 mg/mL, and (C & F) were obtained from LSPMbs and SonoVue of 0.4 mg/mL. (G) was obtained from degas deionized water, and (H) was obtained from air. The echogenic intensity decreased with decreasing concentrations of LSPMbs, and degas deionized water; the echogenic intensity was strong at the interface between the air in the tube and the outside water, and the echogenicity in the tube was attenuated.

Comparisons of measurements of enhanced ultrasound imaging between different concentrations and agents were as follow: (A) vs. D, P = 0.321; (B) vs. E, P = 0.472; (C) vs. F, P = 0.428; and (A) vs. G, A vs. H, B vs. H, G vs. H, A vs. B, A vs. C, B vs. C, D vs. E, and D vs. F, all P < 0.05.

Assessment of affinity of LSPMbs to the liver cancer cells

The fluorescence on the cellular membrane of Huh-7 cells was intensive (Fig. 6), indicating that there was high GPC3 expression. Cells and DiI labled LSPMbs and controls obtained from light microscope (Figs. 7A, 7E and 7I). Cell nucleus of Huh-7 cells presented blue after DAPI staining and being incubated with DiI labled LSPMbs (Figs. 7B, 7F and 7J). LSPMbs with DiI staining combined with the membrane of Huh-7 cells presented red color fluorescence on image (Figs. 7C and 7D); LSGMbs presented a very similar appearance (Figs. 7G and 7H); and LSPMbs with DiI staining after synthesized peptide blocking previously did not show red color fluorescence on the membrane of Huh-7 cells, indicating that LSPMbs had not combined with the membrane of Huh-7 cells (Figs. 7K and 7L).

Figure 6 Image of GPC3 expression of Huh-7 cells obtained by confocal laser scanning microscope.

The fluorescence on the cellular membrane appears intensive.

Figure 7 Images of Huh-7 cells incubated with DiI lablled LSPMbs and controls obtained by light microscope and confocal laser scanning microscope.

(A, E and I) Cells and DiI lablled LSPMbs and controls obtained from light microscope. (B, F and J) Cell nucleus of Huh-7 cells presented blue after DAPI staining and being incubated with DiI lablled LSPMbs. (C and D) LSPMbs with DiI staining combined with the membrane of Huh-7 cells presented red color fluorescence on image. (G and H) LSGMbs presented a very similar appearance. (K and L) LSPMbs with DiI staining after blocked by synthesized peptide did not show red color fluorescence on the membrane of Huh-7 cells, indicating that LSPMbs had not combined with the membrane of Huh-7 cells.

Assessment of LSPMbs in vivo

Enhanced ultrasound imaging assessment of LSPMbs

All xenograft tumors in the mice of the five groups presented a complex of isoechogenicity, hypoechogenicity and anechogenicity (A), E, I, M and Q). Of the study group, after administration of LSPMbs suspension, Huh-7 xenograft tumor presented hyper-enhancement in periphery and hypo-enhancement in center (necrosis) at two seconds (Fig. 8B); the tumor enhancement lasted over 20 s with little change; at 60 s, the tumor still presented hyper-enhancement in periphery and hypo-enhancement in center (necrosis) (Fig. 8C); at 10 min, the tumor presented iso-enhancement with central hypo-enhancement (necrosis) (Fig. 8D). Of the four control groups, the mice injected with LSGMbs, LSPMbs (blocked with GPC3 antibody or synthesized peptide previously), and SonoVue, respectively, presented similar enhancement patterns and sustain time as those in Huh-7 xenograft tumors (Figs. 8F–8H, Figs. 8J–8L, and Figs. 8N–8P), and there was no significant difference; the xenograft tumors presented similar enhancement patterns and sustaining time; of the mice in the group injected with LS suspension, the xenograft tumors did not present enhancement at 2, 20, 60 s, and 10 min (Figs. 8R–8T).

Figure 8 Row 1 is a study group, and rows 2–5 are control groups.

(A, E, I, M and Q) Images of all xenograft tumors presented a complex of isoechogenicity, hypoechogenicity and anechogenicity obtained by convention ultrasound. (B of LSGMbs) After administration of LSPMbs suspension, tumors presented hyper-enhancement in periphery and hypo-enhancement in center (necrosis) at two seconds. (C of LSGMbs) At 60 s, the tumor still presented hyper-enhancement in periphery and hypo-enhancement in center (necrosis). (D of LSGMbs) At 10 min, the tumor presented iso-enhancement with central hypo-enhancement (necrosis). (F, G and H of LSGMbs) and (J, K and L of LSPMbs, after GPC3 blocking) and they presented similar enhancement patterns and sustain time as those in Huh-7 xenograft tumors, and there were no appreciable difference. (N, O and P of SonoVue) The xenograft tumors presented similar enhancement patterns and sustaining time as those using LSPMbs after administration of SonoVue suspension. (R, S and T of LS) The xenograft tumors did not present enhancement at 2, 20, and 60 s, and 10 min after administration of LS suspension.

Comparisons of enhanced ultrasound imaging among different agents at times of two seconds, 20 s, and 10 min, scales of images (B, C, and D of LSPMbs, J, K, and L of LSPMbs after synthesized peptide or GPC3 blocking, F, G, and H of LSGMbs, and N, O, and P of SonoVue) all had no significant difference (all P >  0.05), and there were significant difference between the above scales of images and scales of LS images (all P < 0.001). These indicate that LSPMbs has good capability in CEUS imaging, but the experiments of it did not show targeted imaging in vivo.

Fluorescent imaging assessment of LSPMbs

The fluorescent signal could be visualized all over the body soon after the administration of Cy5.5 labelled LSPMbs suspension. Images acquired at one minute (Fig. 9A), six hours (Fig. 9B) after the initial fluorescent imaging, the fluorescent signal intensity in the tumor area has no significant difference from other areas of the body expect the liver and spleen. The fluorescent signal intensity in the liver and spleen area of the mice was marked stronger than other areas, and the fluorescent signal intensity was similar in three times. The fluorescent signal could not be visualized in the mice 24 h after injection (Fig. 9C). Experiment carried out in the mice blocked previously by injection of GPC3 antibody or synthesized peptide presented the same fluorescent imaging characteristics as those in the mice with Huh-7 xenograft tumors (Figs. 9D–9F). 24 h after intravenous injection of Cy5.5 labelled LSPMbs, four mice with Huh-7 cell xenograft tumors of two in each groups were sacrificed, the tumors and visceral organs were assessed, there were fluorescence in the lungs and liver, and no fluorescence in the tumor and the heart, spleen and kidneys (Fig. 10). These indicate that the Cy5.5 labelled LSPMbs did not selective accumulated in the xenograft tumor, and the LSPMbs did not present detectable targeting ability to the tumor. The reason that the fluorescent signal intensity in the liver and spleen was higher than other areas is believed that the liver and spleen have abundant capillaries and macrophage cells, the macrophage cells can engulf the liposomes, so liposomes in these regions are richer than other regions. The more Cy5.5 labelled LSPMbs aggregated, the stronger the fluorescent signal intensity.

Figure 9 A was acquired at one minute, and B was acquired six hours after the initial fluorescent imaging, the fluorescent signal intensity in the tumor area has no significant difference from other areas of the body expect the liver and spleen. C was acquired 24 h after injection, the fluorescent signal could not be visualized in the mice. D, E and F were acquired from experiments of mice with Huh-7 xenograft tumors blocked using GPC3 antibody presented the same fluorescent imaging characteristics as those in the mice with Huh-7 xenograft tumors.

Figure 10 Fluorescence in the organs.

There were fluorescence in the lungs and liver, and no fluorescence in the tumor, heart, spleen, and kidneys 24 h after intravenous injection of Cy5.5 labelled LSPMbs, after animal sacrificed.

Discussion

LSPMbs presented similar enhancement effect at the same concentration as that the SonoVue performed in vitro and vivo, indicating that the LSPMbs has good capability of enhancement imaging. Optical imaging in vivo using fluorescence and bioluminescence has high sensitivity

and resolution (Imamura, Saitou & Kawakami, 2018). The near infrared dye Cy5.5 allows a fluorescent imaging of deep tissue in rodent animals with low background, providing a possibility of evaluation of the molecular imaging agents. In this study, LSPMbs were labelled with Cy5.5 for targeting GPC3 imaging evaluation, the results showed that they had not presented aggregated fluorescence imaging, indicating that LSPMbs were not targeting retained in the tumor. LSPMbs did not present targeted imaging both at ultrasound imaging and optical imaging.

Unlike iodinated and gadolinium contrast agents for X-ray, CT and MRI that can enter extravascular tissue, common UCAs are confined to the blood pool when administered intravenously, which are consist of microbubbles in suspension which strongly interplay with the ultrasound beam and are readily detectable by ultrasound imaging systems (Chong, Papadopoulou & Dayton, 2018). Molecularly targeted UCAs are created by conjugating the microbubble shell with a peptide, antibody, or other ligand designed to target an endothelial biomarker associated with tumor angiogenesis or inflammation. These microbubbles then accumulate in the microvasculature at target sites where they can be imaged (Chong, Papadopoulou & Dayton, 2018).

Our findings were significantly different from previous reports that vascular endothelial growth factor receptor 2 (VEGFR2) based targeted UCAs (Willmann et al., 2017). These targeted UCAs do not need to extravasate the blood vessels. The VEGFR2 based targeted UCAs can contact and combine with the VEGFR2 on the blood vessels when they enter and flow through the tumor’s vessels, forming focal contrast agent accumulation, and can display focal enhanced imaging at ultrasound scanning. However, our preparation of LSPMbs targeting cellular membranous receptor of the HCC confronts substantial challenge for targeted imaging. Sizes of LSPMbs are 380.9 ±  176.5 nm, which can extravasate the fissure of blood vessel wall of tumor if there are no other impeding factors. But the experimental results did not gain the desired goal. The reasons may be the following factors. The previous study showed that the gap between the epithelia of cancer blood vessel is large enough (380-780 nm) to allow nanoscale materials passing, but the blood vessels contact the cells of tumor and interstitial closely, elevated interstitial fluid pressure in the tumor could restrict convective flow and antibody extravasation, except in large necrosis and hypoxia areas (Wang et al., 2016; Thurber, Schmidt & Wittrup, 2008). Similarly, the targeted LSPMbs needs overcome interface pressure gradient and get enough space to access and bind to the cancer cells of tumor. How the LSPMbs penetrate the blood vessels of tumor, distribute in the tumor and uptake by the cells are difficult to understand or predict (Jain, 1999). A study by Opie showed that the osmotic parameters of tumors (hepatoma, etc.) are much lower than that of normal tissues (Opie, 1949). In this circumstance, the suspension of LSPMbs is harder to be absorbed into tissue by osmotic force of tumor. At tumor sites, the disorganized tumor vascular network, extensively distributed stromal cells (e.g., tumor-associated macrophage, cancer-associated fibroblasts, etc.) and the dense physical barriers of extracellular matrix comprise of the abominable obstacles hampering nanoparticles transport in a tumor (Zhang et al., 2018). The electron microscopy results on a study have confirmed that the opening in tumor extracellular matrix barriers surrounding the cancer cells is generally less than 40 nm (Ng, Lovell & Zheng, 2011). On this condition, LSPMbs of 380.9 ±  176.5 nm are impossible to pass the extracellular matrix opening to access to the cancer cells. A recent study revealed that only 0.7% of systemically administered nanoparticles can reach the tumor sites and less than 14 out of 1 million (0.0014% injected dose) of them are accessed by cancer cells, and that only 2 out of 100 cancer cells interacted with the nanoparticles (Dai et al., 2018). Therefore, if the number and volume of the targeted LSPMbs entering the tumor extravascular part are not enough, it will be impossible to yield visible enhanced imaging effect.

Many researches on extravascular targeted contrast-enhanced ultrasound have been reported in literature, but only a few of them validate in vivo of animals. Mai et al., (2013) reported that a chitosan-vitamin C lipid system had been fabricated and had achieved tumor-selective enhanced ultrasound imaging in a mouse tumor model, but they confirmed only that the fluorescence accumulated highly at the tumor site, other than the nanobubbles. Another extravascular targeted nanobubbles fabricated by Gao et al., (2017) remains to be further verification because of the preliminary results and substantial limitations. Theoretically, if microbubbles enter the tumor interstitials, some liquid solution also enters. There must be enough extravascular space in the tumor to contain and distribute them, only in this condition can the microbubbles in the oscillations of ultrasonic compression and expansion wave generate stronger backscattered acoustic signal and second harmonics for enhanced ultrasound imaging. If many microbubbles are compacted together, their size will be big, and which will generate little backscattered acoustic signal and second harmonics (Sanchez, Varadarajulu & Napoleon, 2009).

Our experimental results, together with earlier published reports by others (Zhang et al., 2018; Ng, Lovell & Zheng, 2011; Dai et al., 2018), strongly suggested that to develop a targeted material for extravascular contrast-enhanced ultrasound imaging, cutting-edge precisive experiments should be conducted firstly to ascertain that the material can penetrate the blood vessel and wade through the cellular matrix and stroma to access and bind to the target cells, and the accommodation space for the materials is adequate. In future, the development of materials for extravascular contrast-enhanced ultrasound imaging may be emphasis that using specific materials such as cell-penetrating peptides, a disulfide-bridged cyclic RGD peptide, named iRGD (internalizing RGD, c(CRGDK/RGPD/EC)), which is a tumor-homing peptide that can bind to avb3 integrin with high affinity and specificity to construct the targeted material. A material integrated iRGD peptide may increase penetration of the blood vessels and matrix, and facilitate accumulation and increase the probability of enhanced imaging (Yan et al., 2017; Cho et al., 2016). Augmentation of enhanced permeability and retention effect of targeted material through using NO-releasing agent such as nitroglycerin or angiotensin-converting enzyme inhibitors, and albumin-protein interactions using S -nitrosated human serum albumin dimer, etc, may increase targeted material accumulation and the probability of enhanced imaging (Maeda, 2012; Kinoshita et al., 2017).

Conclusions

Collectively, a new material of LSPMbs has been prepared, which has good effect of enhanced ultrasound imaging, but it did not exhibit targeted imaging effect in vivo of animal experiments. The causes may be that the volume of LSPMbs passing the tumor blood vessels and entering the tumor parenchyma was very limited, and the LSPMbs cannot pass the fissures of extracellular stroma and matrix surrounding the cancer cells to access and bind to the cancer cells. Therefore, a potential target of GPC3 on hepatocellular carcinoma for extravascular targeted imaging may not be realized in contrast-enhanced ultrasound. The future research should focus on that whether a candidate targeted material for extravascular contrast-enhanced ultrasound imaging can penetrate the blood vessel and wade through the cellular matrix and stroma to access and bind to the target cells, and whether the accommodation space for the materials is adequate.

Supplemental Information

Supplemental Information 1 LSPMbs Size Measurements.

The numbers in the table were the original parameters of sizes of LSPMbs measured using a Zetasizer Nano S90.

Click here for additional data file.

Supplemental Information 2 LSPMbs Zeta Potential Measurements.

The numbers in the table were the original parameters of Zeta potentials of LSPMbs measured using a Zetasizer Nano S90.

Click here for additional data file.

Supplemental Information 3 MTT Assay for Cytotoxicity: Representative Measurement of Optical Density Value

Experiments for cytotoxicity were performed by measurement of optical density value; the experiments were performed many times, one measurement was selected as a representative, and the numbers in the table were the original parameters obtained at 72 h.

Click here for additional data file.

Supplemental Information 4 Cell Index for Assay of Cytotoxicity.

The numbers in the table were the curated data of measurement of optical density value in the MTT assay obtained from measurements at different times.

Click here for additional data file.

Supplemental Information 5 Evaluation of Enhanced Ultrasound Imaging of LSPMbs in Vitro

The enhancement effects of different agents in vitro were determined using Photoshop software (Adobe Photoshop CS3, Adobe Systems Inc, CA, USA). Parameters in A and D were obtained from LSPMbs and SonoVue suspension of 1.6 mg/mL, respectively; B and E were obtained from 0.8 mg/mL; C and F were obtained from 0.4 mg/mL; G was obtained from degas deionized water; and H was obtained from air.

Click here for additional data file.

Supplemental Information 6 Assessment of LSPMbs in Vivo: Enhanced Ultrasound Imaging of LSPMbs

The enhancement effects of different agents in vivo were determined using Photoshop software (Adobe Photoshop CS3, Adobe Systems Inc, CA, USA). Parameters of A, E, I, M and Q were of xenograft tumors in the mice of the five groups before administration of experimental agent; parameters of B, C and D were of measurements at two seconds, 60 s, and 10 min after administration of LSPMbs suspension, respectively; parameters of F, G and H were of measurements at two seconds, 60 s, and 10 min after administration of LSGMbs suspension, respectively; parameters of J, K and L were of measurements at two seconds, 60 s, and 10 min after administration of LSPMbs (blocked with GPC3 antibody or synthesized peptide previously) suspension, respectively; parameters of N, O and P were of measurements at two seconds, 60 s, and 10 min after administration of SonoVue suspension, respectively; parameters of R, S and T were of measurements at two seconds, 60 s, and 10 min after administration of LS suspension, respectively.

Click here for additional data file.

Supplemental Information 7 Raw materials.

Images obtained during this study.

Click here for additional data file.

Additional Information and Declarations

Competing Interests

Author Contributions

Animal Ethics

Data Availability

The authors declare there are no competing interests.

Size Wu conceived and designed the experiments, performed the experiments, analyzed the data, prepared figures and/or tables, authored or reviewed drafts of the paper, and approved the final draft.

Xiyuan Lin performed the experiments, analyzed the data, prepared figures and/or tables, and approved the final draft.

The following information was supplied relating to ethical approvals (i.e., approving body and any reference numbers):

Institutional Animal Care and Use Committee of Hainan Medical University approved the study (2018-02-27).

The following information was supplied regarding data availability:

The raw measurements are available as a Supplemental File.

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
