# Peer review of "Trials in developing a nanoscale material for extravascular contrast-enhanced ultrasound targeting hepatocellular carcinoma"

_PeerJ, doi:10.7717/peerj.10403_

## Round 0.1 · original submission · Major Revisions

Please read carefully the following comments by two reviewers who are experts in this research areas. I look forward to receiving your revised version of this manuscript.

Reviewer 1 ·

Basic reporting

1. This article is to develop the image enhanced materials for the ultrasound examination especially specific tumor targeting such as hepatocellular carcinoma.
2. The article is written in English using clear and unambiguous text and complies with the professional standards of courtesy and expression.
3. This Article is reviewed at the standpoint of an oncology clinician and the experimental pathology of malignant tumors.

Experimental design

1. This experimental study was executed in the adequate animals evaluation, data processing including statistical analyses scientifically and ethically.
2. Materials and Methods, Line 70. In this study, there is only one image intensifier material, and there is no description of the selection from multiple candidates during the development process.
3. Materials and Methods, Line 71. There is no description about the reason for selecting the gas(SF6) to be enclosed in the microbubbles.
4. Materials and Methods, Line 72. There is no detailed description of GPC3 antibody and peptides that bind to the microbubbles.
5. Materials and Method,s Line 130. There is no description of the experiment on the particle size of liposome microbubbles filled with SF6, considering only the change in the solution concentration.

Validity of the findings

The following issues need to be fixed.
1. Figure 4. The concentration in the right column is unknown. After 72 hours, there is a slight decrease in the Cell Index. In the safety evaluation, it is necessary to extend the observation period.
2. Figure 5-8. Please insert a size bar on each image.
3. Figure 6. The explanation of “The fluorescence on the cellular membrane appears stronger than that in cellular plasma. ”. This findings cannot be recognized in the image.
4. Figures 7. Fluorescence images cannot be recognized in photographs. It is better to replace images with Supplement Figure images.
5. Figure 8, There is no explanation for 3rd row. And the initial elapsed time is missing in this figure.
6. Figure 10. In the safety evaluation of systemic administration (intravenous administration), evaluations of the pulmonary embolus are necessary. A histopathological image of the organs including the lungs should be presented.
7. Supplement Figure. Image of xenograft tumor of the mouse obtained by duplex ultrasound with a scanner of frequency of 12 MHz
8. HCC tumors are rich in blood flow. However, the Doppler echo image of the xenograft tumor in the Supplement Figure does not show a blood flow signal. There are problems in selecting the xenograft tumor.

Additional comments

1. It has little advantage over existing intravenous image intensifier materials for ultrasonography.
2. Since this is negative final results, it is necessary to describe details above-mentioned problems for the future developments.
3. As a result of the peer review, a major revision is required.

Reviewer 2 ·

Basic reporting

N/A

Experimental design

N/A

Validity of the findings

N/A

Additional comments

Medical imaging is an important approach for the diagnosis of HCC, and there is still scope for improvement to achieve accurate diagnosis. Therefore, what we need are novel and effective diagnostic tools with "better" abilities.
In this manuscript, Wu and Lin aimed to formulate a material that can combine with GPC3 of HCC for targeted enhanced ultrasound. A material of SF6 filled liposome microbubbles and conjugated with LSPMbs was prepared and assessed. Liposome microbubbles were made of DPPC, DPPG, DSPE-PEG2000, and SF6, followed filling SF6, and conjugating peptide. LSPMbs showed high affinity to Huh-7 cells in vitro, presented good enhanced ultrasound effects, did not show cytotoxicity. However, unfortunately they did not exhibit targeted fluorescence and enhanced ultrasound effects in xenograft tumors.
As the authors noted, the material described in this manuscript appears not to be utilized in clinical setting. Furthermore, the authors did not clearly show the reason of an unsuccessful attempt. In this point, the authors discussed it based on previous literature that suggest small amount of LSPMBs of 380 +/- 176.5 nm cannnot access to HCC cells.
This reviewer fully respects author’s efforts and the results of this experiment, though still wonders the fundamental meaning of this paper.

---

## Round 0.2 · Minor Revisions

I have received the review result from a reviewer who has recommended that your manuscript would be acceptable. However, the reviewer is also concerned about the title of your manuscript including the term of "negative outcome", which would reduce the value of the manuscript. Therefore, I would suggest your revising the title to convey a more neutral impression.

I would suggest "Trials in developing a nanoscale material for extravascular contrast-enhanced ultrasound targeting hepatocellular carcinoma"

Reviewer 1 ·

Basic reporting

1. This article is to develop an image enhanced material for the ultrasound examination targeting specific tumor such as HCC.
2. This Article is re-reviewed at the standpoint of an oncology clinician and the experimental pathology of malignant tumors.

Experimental design

1. This experimental study was executed in an adequate animal evaluation, data processing including statistical analyses scientifically and ethically.
2. The response for each reviewers’ comments seems to be rewritten adequately and precisely.

Validity of the findings

1. The author carefully revised Figures according to reviewer’s comments.

Additional comments

1. Reviewers have already pointed out the poor experimental results. And article authors are also aware of this issue.
2. Although final results is negative, this article details the process of development materials for extravascular ultrasound targeting.
3. The title "negative outcome" might lose the value of the paper, so it is better to rewrite the title in a different expression.
4. A result of the peer review, acceptable except the title expression.

---

## Round 0.3 · accepted · Accept

I appreciate your revising the manuscript, and I am so happy that your manuscript has been accepted for publication in PeerJ. I hope that you will submit future manuscripts to PeerJ.